# Safety of Dietary Camelina Oil Supplementation in Healthy, Adult Dogs

**DOI:** 10.3390/ani11092603

**Published:** 2021-09-05

**Authors:** Scarlett Burron, Taylor Richards, Keely Patterson, Caitlin Grant, Nadeem Akhtar, Luciano Trevizan, Wendy Pearson, Anna Kate Shoveller

**Affiliations:** 1Department of Animal Biosciences, University of Guelph, Guelph, ON N1G 2W1, Canada; sburron@uoguelph.ca (S.B.); tricha16@uoguelph.ca (T.R.); kpatte10@uoguelph.ca (K.P.); akhtarn@uoguelph.ca (N.A.); wpearson@uoguelph.ca (W.P.); 2Department of Clinical Studies, University of Guelph, Guelph, ON N1G 2W1, Canada; grantc@uoguelph.ca; 3Departamento de Zootecnia, Universidade Federal do Rio Grande do Sul, Porto Alegre 91540-000, Rio Grande do Sul, Brazil; ltrevizan@ufrgs.br

**Keywords:** omega-3, omega-6, fatty acids, camelina oil, flaxseed oil, canola oil, canine nutrition

## Abstract

**Simple Summary:**

Dietary sources of omega-6 and omega-3 fatty acids are essential in canine diets and provide many health benefits. Camelina (*Camelina sativa*) is a low-input, high-yield oilseed crop that produces highly unsaturated oil (~90%), has a desirable omega-6 to omega-3 fatty acid ratio, and high concentrations of tocopherols. These attributes make camelina oil a potential alternative to other plant-based oil products for canine nutrition. In the current study, we evaluated the safety of dietary camelina oil supplementation in dogs over a 16-week period in dogs by assessing body weight, body condition score, food intake, and hematology and biochemistry analytes. Differences in the results were minimal compared to dogs fed canola and flaxseed oil, which are regarded as safe for use in canine diets. Therefore, camelina oil can be considered safe for use in the nutrition of adult dogs.

**Abstract:**

This study aimed to determine whether camelina oil is safe for use in canine diets, using canola oil and flax oil as controls, as they are similar and generally regarded as safe (GRAS) for canine diets. A total of thirty privately-owned adult dogs of various breeds (17 females; 13 males), with an average age of 7.2 ± 3.1 years (mean ± SD) and a body weight (BW) of 27.4 ± 14.0 kg were used. After a 4-week wash-in period using sunflower oil and kibble, the dogs were blocked by breed, age, and size and were randomly allocated to one of three treatment oils (camelina (CAM), flax (FLX), or canola (OLA)) at a level of 8.2 g oil/100 g total dietary intake. Body condition score (BCS), BW, food intake (FI), and hematological and select biochemical parameters were measured at various timepoints over a 16-week feeding period. All of the data were analyzed with ANOVA using the PROC GLIMMIX of SAS. No biologically significant differences were seen between the treatment groups in terms of BW, BCS, FI, and hematological and biochemical results. Statistically significant differences noted among some serum biochemical results were considered small and were due to normal biological variation. These results support the conclusion that camelina oil is safe for use in canine nutrition.

## 1. Introduction

Omega-6 linoleic acid (C18:2n-6; LA) and omega-3 alpha-linolenic (C18:3n-3; ALA) fatty acids are essential in canine diets, as dogs are not able to produce these fatty acids endogenously [1]. Omega-3 (n-3) fatty acids in particular have been shown to have many health-promoting outcomes, including cardioprotective effects [2,3,4], anti-inflammatory and immune modulating benefits [5,6,7], and improved skin and coat health properties [8,9]. The ideal n-6:n-3 fatty acid ratio for canine diets is between 5:1 and 10:1, and including n-3 rich ingredients is typically needed to achieve this desired ratio when formulating canine diets [7]. Fish oils are a commonly used source of n-3 supplementation due to their high levels of eicosapentaenoic acid (EPA) and docosahexaenoic acid (DHA); however, the large-scale use of fish oil in canine diets is not an environmentally sustainable option long-term, leaving a need for plant-based oil alternatives [10,11,12]. Canola, corn, soybean, and sunflower oil are commonly used plant-based lipid sources in the pet food industry; however, these oils have much higher levels of omega-6 (n-6) than n-3 fatty acids, with n-6:n-3 ratios of 1:0.59, 1:0.01, 1:0.12, and 1:0.00, respectively [12,13]. Flax oil does have an n-6:n-3 ratio (1:4.19) that is favourable in bringing canine diets to a desirable n-6:n-3 ratio, though the continuous cropping of flaxseed oil plants is rare in North America due to their sensitivity to winter climates and diseases or pests [12,14]. This leaves room in the market for another plant-based oil that can be easily cultivated and that can provide high n-3 inclusion while still being economically and environmentally sustainable.

Camelina (*Camelina sativa*), also known as false flax or gold of pleasure, is an oilseed plant of the Brassicaceae (mustard) family that can be grown in a variety of climates, seasons, and soil types due to its short growing season and tolerance to drought and low temperatures [15,16,17,18]. Camelina is known to be a low-input, high-yield crop that can perform favourably in poor soils and that is resistant to many pests that affect other oilseed crops [19,20]. The camelina oilseed has a high oil yield (~40% oil and ~60% meal), and the resulting oil is highly unsaturated (~90%), has a desirable n-6:n-3 ratio (1:1.8), and contains high concentrations of tocopherols [21]. Camelina meal has been approved as safe for use in broiler chickens, cattle fed in confinement for slaughter, and laying hens at an inclusion of up to 10% of the diet, and the use of camelina oil in fish feed has also been approved [12,22].

The objective of this study was to determine the safety of camelina oil on canine health by comparing it to flaxseed oil (favourably high in n-3 fatty acids) and canola oil (commonly used in pet foods). Since camelina meal has already been approved as safe for use in many livestock species and since flaxseed oil and canola oil are currently used and regarded as safe for use in canine diets, we hypothesized that there would be no negative effects of camelina oil supplementation in adult dogs on food intake (FI), body weight (BW), body condition score (BCS), and hematology and serum biochemistry analytes. 

## 2. Materials and Methods

### 2.1. Animals, Health Assessment, and Housing

The present study was approved by the University of Guelph’s Animal Care Committee (Animal Use Protocol #4365) and was in accordance with national and institutional guidelines for the care and use of animals. A total of thirty privately-owned, adult dogs of mixed breeds (17 females: 16 spayed, 1 intact; 13 males: 10 neutered, 3 intact), with an average age of 7.2 ± 3.1 years (mean ± standard deviation, SD) and a BW of 27.4 ± 14.0 were recruited to participate in the study. All of the dogs met the following inclusion criteria: clinically healthy on assessment, showing no abnormalities on routine biochemistry and complete blood count (CBC) blood tests, and having no known dietary allergies or skin conditions before being accepted to the study. The dogs were housed at the owner’s homes where they followed their usual daily routines. 

### 2.2. Diets, Study Design, and Food Intake

All of the dogs were acclimated to a dry extruded commercial kibble (SUMMIT Three Meat Reduced Calorie Recipe, Petcurean, Chilliwack, BC, Canada) (Table 1), sunflower oil (SA Kernel-Trade, Kuiv, Ukraine) (Table 2), and treats (proximate analysis: metabolizable energy 3039 kcal/kg; crude protein minimum 65%; crude fat minimum 5.1%; crude fibre maximum 4.0%; moisture max 9.56%) (Beef Tendersticks, The Crump Group, Brampton, ON, Canada) over a 4-week wash-in period. During the wash-in period and throughout the study, the daily portions of food, oil, and treats were pre-weighed by the researchers and were given to owners in two-week intervals to be offered to the dogs daily at a frequency determined by the owners. In order to avoid the occurrence of lipid peroxidation, the oil was mixed with the food immediately before feeding. Leftover food was returned to the researchers and was subsequently weighed and recorded to calculate the FI. 

The dogs were initially fed to meet their maintenance energy requirements (110 kcal *×* BW(kg)^0.75^); then, every two weeks, their BW and BCS were recorded, and the amount of feed was adjusted to maintain BW and BCS throughout the study. Oil was included in the diet at a level of 8.2 g of oil per 100 g of total intake, bringing the total dietary lipid content to 20% inclusion on an as-fed basis. Treats were included in the diet up to 2.5 g per 100 g total intake, and the remaining proportion of the diet was provided as kibble.

A randomized complete block design (RCBD) with repeated measures was used for this study. After the wash-in period, the dogs were blocked by breed, age, and size before being randomly allocated to one of three treatment diets: camelina oil (CAM) (*n* = 10; 8 females; 2 males), flax oil (FLX) (*n* = 10; 5 females; 5 males), or canola oil (OLA) (*n* = 10; 4 females; 6 males). Sunflower oil was replaced with either CAM, FLX, or OLA, and feeding continued as described for 16 weeks.

### 2.3. Body Weight and Body Condition Scores

Body weight was measured every two weeks using a Redmon Precision Digital Pet Scale (Redmon Co., Peru, IN, USA) and BW was then used to calculate dietary food, oil, and treat intake. Body weights on weeks 0, 2, 4, 10, and 16 were measured after an overnight fast and at the same time of day. This allowed the researchers to eliminate variability due to time since last feeding and diurnal effects on BW. For each dog, BCS was assessed on weeks 0, 2, 4, 10, and 16 using a validated 9-point scale [24].

### 2.4. Sample Collection and Analysis

The dogs were fasted for a minimum of 10 h, and blood samples were collected via cephalic venipuncture using a syringe (Becton, Dickinson and Company, Franklin Lakes, NJ, USA). Of the collected blood, 1mL was put into a K2 EDTA 10.8 mg Vacutainer (Becton, Dickinson and Company, Franklin Lakes, NJ, USA) for the hematological indices, and 1 mL was put into a serum vacutainer (Becton, Dickinson and Company, Franklin Lakes, NJ, USA). Fasted blood samples were collected within the 2 weeks prior (pre-study or baseline) to the start of the wash-in diet with sunflower seed oil supplementation, and on weeks 4, 10, and 16 after starting the diets supplemented with treatment oils. Collection was taken at these timepoints to allow the researchers the opportunity to assess the effects of both the diet and the treatment oils on standard veterinary diagnostic measures and markers of health and nutritional status using serum and whole blood. 

For hematology (complete blood cell count; CBC), values were determined by the Animal Health Laboratory (AHL) (University of Guelph, Guelph, Canada) using an Advia 2120 hematology analyzer (Siemens Global, Munich, Germany). EDTA samples were stored on ice and were analyzed on the same day, or, in a few cases where the laboratory was closed, blood smears were performed and stored at room temperature, and EDTA samples were refrigerated until analyses were available. Hematological samples were analyzed for blood leukocyte count (WBC), erythrocyte count (RBC), hemoglobin (Hb), hematocrit (Hct) (RBC × MCV), mean cell volume (MCV), mean cell hemoglobin (MCH) (Hb/RBC), mean corpuscular hemoglobin concentration (MCHC) (Hb/Hct), red cell distribution width (RDW), platelet count, mean platelet volume (MPV), plateletecrit, and total solid (T.S.) protein as well as segmented neutrophil, lymphocyte, monocyte, and eosinophil counts.

For the biochemical analysis, blood from the serum vacutainer (1 mL) was allowed to clot and was centrifuged at 7200× *g* for 15 min using an accuSpin Micro 17 centrifuge (Thermo Fisher Scientific, Waltham, MA, USA). Then, the serum aliquots were collected and were analyzed on same day or were frozen at −80 °C until analysis. Serum samples were analyzed for calcium, phosphorus, magnesium, sodium, potassium, chloride, carbon dioxide, anion gap, sodium:potassium (Na:K) ratio, total protein, albumin, globulin, albumin:globulin (A:G) ratio, urea, creatine, glucose, cholesterol, total bilirubin, conjugated bilirubin, free bilirubin, alkaline phosphatase (ALP), steroid-induced ALP, gamma-glutamyl transferase (GGT), alanine aminotransferase (ALT), creatine kinase (CK), amylase, lipase, and calculated osmolarity using a cobas 6000 c501 analyzer (Roche Diagnostics Internation AG, Rotkreuz, Switzerland).

The biomarker reference intervals used by AHL for both hematology and biochemistry have been previously determined and used 86 healthy, fasted adult dogs of various breeds, lifestyles, and life stages. 

### 2.5. Statistical Analysis

All of the statistical analyses were performed using the PROC GLIMMIX of SAS Studio^®^ software (v.9.4., SAS Institute Inc., Cary, NC, USA). Dog was the experimental unit, and treatment oils and sex were treated as fixed effects. Week was treated as a repeated measure. An analysis of variance (ANOVA) was performed to assess the effects of treatment on BW, BCS, FI, and biochemistry and hematology analytes. Assumptions of residuals for all parameters were assessed using the Shapiro–Wilk to test normality. Residuals were not uniformly distributed for ALP, ALT, CK, and lipase, and as such, data were log-transformed prior to analysis. Least-square means were used to assess differences in the means of treatment, week, and treatment by week interactions. When the fixed effects were significant, the means were separated using Tukey–Kramer adjustments. Significance was declared at a *p* ≤ 0.05.

## 3. Results

Due to difficulty with blood collection, no hematology or biochemistry samples were obtained from Dog #21 on week 16. Dog #14 was put on antibiotics for a urinary tract infection before week 16, and as a result, CBC and biochemistry data were excluded to maintain consistency between all dogs. Dog #10 dropped out of the study after week 4 for unrelated medical reasons. Further, only partial samples were obtained on five occasions (Dog #1, week 1; Dog #9, week 16; Dog #13, week 4 and week 10; and Dog #18, week 4), leading to some missing values in the CBC and biochemistry data. All of the aforementioned data were excluded, and statistical analysis was performed using PROC GLIMMIX in order to account for the missing data points and the subsequent unequal number of observations between groups. A total of two additional animals were included in each treatment group to account for possible variation arising from differences in breed, age, and sex of the dogs as well as to account for the possibility of a dog being removed or dropping out of the study or for the possibility of missing data points. Statistically significant differences are outlined below, but the statistical analysis mostly found very small differences between the treatment groups and the values within the given reference ranges and therefore were determined to be biologically insignificant. 

### 3.1. Body Weight, Body-Condition Score, and Food Intake

In order to maintain the BW, researchers decreased FI after week 2 (FI = 12.13 g/kg/day) and continued to decrease FI until week 16 (FI = 10.66 g/kg/day) as needed based on the bi-weekly BW measurements of each dog (Table 3). Body weight and BCS were similar among treatments and across weeks (*p* > 0.05). Food intake did not differ among treatment or treatment by week interactions (*p* > 0.05); however, FI did differ across weeks (*p* < 0.0001). Specifically, FI was greater at week 2 than at weeks 10, 12, and 14; FI at pre-study and at weeks 4 and 8 was greater than at weeks 6, 10, 12, and 14, and FI at weeks 6, 10, and 12 was greater at than week 14 (*p* < 0.0001; Table 3).

### 3.2. Hematology

All hematology outcomes are listed in Table 4. No hematological biomarkers differed among the treatment groups (*p* > 0.05); however, pooled data for MCH, total soluble protein, and lymphocyte count differed across weeks (*p* < 0.05). MCH was greater at week 16 than at week 10, but no differences were seen from pre-study or week 4 (*p* = 0.0121). Total soluble protein was greater at pre-study than at week 4, but it was not different at weeks 10 and 16 (*p* = 0.0002), and lymphocyte count was greater at pre-study than at weeks 4, 10, and 16 (*p* = 0.0076). All mean estimates stayed within the AHL hematology reference range, except for T.S. protein, which had estimates that were 7% above the reference range.

### 3.3. Biochemistry

All of the biochemistry outcomes are listed in Table 5. The majority of biochemical biomarkers did not differ among treatment groups (*p* > 0.05); however, statistically significant differences among treatments were found for GGT and ALT. GGT was greater in the CAM and FLX treatment groups than in the OLA group (*p* = 0.0130). ALT was greater for the CAM than FLX treatment groups, but neither were different from the OLA group (*p* = 0.0209). 

For treatment by week interactions, conjugated bilirubin was greater at week 4 than at pre-study for the CAM treatment group, but no difference was seen with any other treatment by week interactions (*p* = 0.0208). No other differences were seen for treatment by week interactions for the biochemical analytes (*p* > 0.05). 

When the treatment groups were pooled, calcium, phosphorus, magnesium, creatine, free bilirubin, and amylase did not differ across weeks (*p* > 0.05); however, all of the other biochemical biomarkers showed differences across weeks (*p* < 0.05). Values were greater at pre-study than at week 4 for CO_2_ (*p* = 0.0025), urea (*p* = 0.012), and lipase, but these values not different at weeks 10 and 16 (*p* = 0.0182). Values at weeks 4, 10, and 16 were greater than the pre-study values for total bilirubin (*p* = 0.0006), ALP (*p* < 0.0001), and ALT (*p* = 0.0005). Values at week 16 were greater than the pre-study values for CK (*p* = 0.0290), glucose (*p* = 0.0245), chloride (*p* = 0.0045), and conjugated bilirubin, but these values were not different at weeks 4 or 10 (*p* = 0.0354). Sodium levels were greater at weeks 4 and 16 than they were at week 10 but were not different from the pre-study values (*p* < 0.0001). Potassium levels were greater at pre-study than they were at weeks 10 and 16, and the values at week 4 and week 16 were greater than the values at week 10 (*p* = 0.0002). The anion gap values were greater than the pre-study values at weeks 4 and 16, but these values were not different from those at week 10 (*p* < 0.0001). The Na:K ratio was greater than the pre-study Na:K ratio at weeks 10 and 16, but it was not different from the Na:K ratio at week 4 (*p* = 0.0015). Total protein was greater at week 10 than it was at week 4, but it was not different from the pre-study values or the values at week 16 (*p* = 0.0086). Albumin values were greater at week 16 than they were at weeks 4 and 10, and no difference was seen from the pre-study values (*p* = 0.0078). Globulin values were greater at week 4 than they were at weeks 10 and 16 but were not different than the pre-study values (*p* = 0.0058). The A:G ratio was greater at week 16 than at week 4, but it was not different from the pre-study A:G ratio or the A:G ratio at week 10 (*p* = 0.0049). Values were greater at week 4 than the pre-study values for steroid-induced ALP (*p* = 0.0112) and amylase (*p* = 0.0410) but were not different from those at weeks 10 or 16. Calculated osmolarity was greater at week 10 than it was at all other time points (*p* < 0.0001). 

All mean estimates for treatment, week, and treatment by week interactions stayed within the AHL reference range for all of the biochemistry biomarkers. It is worth noting, however, that while all means estimates for ALP stayed within the reference range, the raw data showed that four dogs were below the reference range at one or more time points throughout the study, and six dogs exceeded the upper limit of the reference range by at least 25% at one or more time points throughout the study. A urinalysis was performed for all dogs exceeding the reference range by more than 25% in order to assess whether there were any abnormalities with the concentrating abilities of their kidneys or any proteinuria that could be indicative of the ALP increase being more than just benign; however, no abnormalities were noted in the urinalysis for any of the dogs.

## 4. Discussion

The purpose of this study was to determine the safety of dietary camelina oil supplementation on canine health by comparing dogs fed camelina oil to dogs fed flaxseed oil or canola oil in order to obtain GRAS certification and an American Feed Control Officials (AAFCO) ingredient definition for camelina oil. Minimal differences were observed in BW, BCS, FI, and hematological and serum biochemical profiles over a 112-day period for dogs fed supplemental camelina oil compared to the control treatment oils, flax and canola. 

### 4.1. BW, BCS, and Food Intake

In order to maintain BW and BCS, FI was adjusted throughout the course of the study and was reduced by 11.2% from baseline (week 0) to week 16. The initial amount of food offered was calculated for all dogs using the same metabolic energy requirement coefficient, regardless of age, sex, or neuter status. Variability based on the aforementioned factors was likely a large contributor to why food intake had to be restricted throughout the course of the study, as it has been shown that estimating the maintenance energy requirements of pet dogs based on BW alone is not always as accurate as when husbandry, neuter status, age, sex, and activity levels are factored into the estimation [25]. While this impacted the statistical results for BW, BCS, and FI—and perhaps other variables—the differences between the treatment group results for all of the outcome variables that were measured were small and were determined to be biologically insignificant when compared to normal intervals. Alternatively, if the dogs were allowed to gain or lose weight, then we would have had to account for that as a potential independent variable that could affect all of the parameters that we evaluated.

### 4.2. Hematology and Biochemistry

Current data in dogs are contradictory as to whether high-fat diets are associated with an increase or decrease in serum cholesterol concentrations; however, fat type, rather than total crude fat inclusion, may play a larger role in serum cholesterol concentrations [26,27,28,29]. In the present study, there was a significant change in the cholesterol concentrations for the treatment and week interaction, where OLA was significantly higher at week 10 than it was for the pre-study values and where FLX was significantly lower at week 10 than it was for the pre-study values; however, no significance was seen among treatments or weeks, and all of the values remained within the reference range (Animal Health Laboratory, Guelph, ON, Canada). This indicates that neither the high dietary lipid inclusion nor the specific lipid source resulted in a change in the serum cholesterol concentrations of the dogs overall in the present study.

Canine serum ALP activity is a non-specific biomarker for cholestasis and can also be increased in young large breed dogs due to bone growth and increased levels of exogenous or endogenous corticosteroids [30]. The mean serum ALP activity results remained within the reference range for all time points and among all treatment levels; however, a statistically significant increase was seen at weeks 4, 10, and 16 for pooled data compared to pre-study values. These results contradict other studies, where dogs fed high-fat diets showed decreased ALP concentrations over time [26,29]. It remains unclear, however, whether the changes to ALP concentrations observed by Swanson et al. (2004) and Anturaniemi et al. (2020) were related to the dietary lipid content or other factors such as the inclusion levels of proteins, carbohydrates, or other nutrients as well as differences in analytical methods. While we did see an increase in the ALP of 25% above the reference range for six dogs and even though four dogs were below the reference range at one or more time point, these observations were independent of treatment oil, and all of the dogs remained clinically healthy with no other abnormalities reported by the owners. Increases in ALP are non-specific, and ALP concentrations may be influenced by a variety of factors, including sex, age, and breed [29,31,32,33,34]; as such, the large variability in both the current data as well as in the literature suggests that the variability in the ALP results are due to normal biological variation or other non-treatment related factors.

While significant differences were seen among treatments for both GGT and ALT, all values remained within reference range (Animal Health Laboratory, Guelph, ON, Canada). To the authors’ knowledge there is presently no conclusive evidence to suggest that dietary composition has a significant effect on either of these liver-related biomarkers in dogs. Swanson et al. (2004) found no significant differences in ALT or GGT concentrations in dogs fed an animal-based diet (20% crude fat) compared to dogs fed a plant-based diet (8% crude fat) [29]. Ober et al. (2016) found no significant differences in ALT in dogs fed a high-fat performance diet, a maintenance diet, or a maintenance diet with supplementary corn oil [35]. Similarly, Anturaniemi et al. (2020) found no differences in ALT in dogs fed a heat-processed high carbohydrate diet versus dogs fed a non-processed high-fat diet [26]. In training sled dogs fed two different diets with a high fat content (53.7% for Diet A and 48.5% for Diet B), an increase in ALT was measured for Diet B at weeks 7 and 20 from week 0, but ALT concentrations decreased from weeks 20 to 24 [36]. Though an increase was seen at certain timepoints for ALT, such changes could also be attributed to differences in the dietary protein content in the diets [36]. For the present study, both the ALT and GGT levels remained at the lower end of the reference range, and no other significant effects among the treatments were seen for other liver related biomarkers, indicating that the dietary supplementation of CAM, when compared to FLX and OLA, had no negative effects on hepatic health over the 16-week period. 

Values for the T.S. protein did not differ among treatments; however, all values were above reference range by up to 7% (Animal Health Laboratory, Guelph, ON, Canada). This is a semi-quantitative measurement of plasma proteins and is much less accurate than the biochemical analysis of serum total protein included in the biochemical panel, which was within normal ranges. Therefore, the minor changes in the T.S. protein above the reference range were considered irrelevant for the safety assessment of CAM.

An increase was seen in total bilirubin by week and conjugated bilirubin for both week and treatment by week interaction; however, all values remained within the reference range, and no differences were observed among treatments. It is difficult to compare bilirubin values of the present study with existing literature in dogs, as reference ranges show variability among different laboratories; therefore, values cannot be accurately compared [26,37]. 

### 4.3. Considerations

This study used client-owned dogs, which provided researchers an opportunity to study the real-world effects of camelina oil on canine health, as opposed to in a laboratory setting. This more accurately depicts how camelina oil will affect the canine population as a whole, as the study participants represent various breeds, ages, sizes, and activity levels. 

The authors also acknowledge possible limitations that could have resulted from working with client-owned dogs that do not live in a controlled environment. Though the owners were asked to report any deviations from the provided study diet, the researchers could not control for any other possible foods that could have been given to the dogs, with or without the owner’s knowledge. Unknown deviations from the diet, the failure of the owners to report any veterinary care, medications, or supplements throughout the study, as well as differences in environments and routines between households could have impacted BW, BCS, FI, and hematology and biochemistry analytes. Since the dogs were brought to campus for blood collection, stress may have had an impact on some concentrations of hematology and biochemistry analytes. Furthermore, the recorded FI could have been impacted by the feeding schedules of owners, failure to return leftover food, or multi-dog households where there is a possibility that the provided food was consumed by other dogs. Finally, because the food was mixed with oil before feeding, researchers were unable to determine oil and food intake separately. Though the dogs were blocked for breed, age, and size, sources of variation due to environmental conditions could not be controlled for in the statistical analysis. 

## 5. Conclusions

This study presents information about the effects of dietary camelina oil supplementation on body weight, body condition score, food intake, and hematology and biochemistry analytes compared to canola and flaxseed oil in healthy, adult dogs. Canola and flaxseed oil are already considered to be safe for use in canine diets, and differences in the aforementioned parameters were minimal among treatment groups. As such, we suggest that camelina oil should be considered safe for use in canine nutrition.

## Figures and Tables

**Table 1 animals-11-02603-t001:** Diet nutrient content of SUMMIT Three Meat Reduced Calorie Recipe on an as-fed basis and ingredient composition ^1^.

Nutrient Contents	Analyzed Content (As Fed Basis)
Moisture (%)	8.00
Crude Protein (%)	23.0
Nitrogen-Free Extract (%) ^2^	52.0
Crude Fibre (%)	2.80
Crude Fat (%)	9.00
Omega 6 (%)	2.00
Omega 3 (%)	0.83
Linoleic Acid (%)	1.90
DHA (%)	0.01
Ash (%)	7.10
Metabolizable Energy (kcal/kg) ^3^	3613

^1^ Chicken meal, whole brown rice, whole white rice, barley, oatmeal, chicken fat (preserved with mixed tocopherols), peas, lamb meal, salmon meal, natural chicken flavour, whole dried egg, sunflower oil, rice bran, flaxseed, dried kelp, dicalcium phosphate, potassium chloride, choline chloride, sodium chloride, calcium carbonate, vitamins (vitamin A supplement, vitamin D3 supplement, vitamin E supplement, niacin, L-ascorbyl-2- polyphosphate (a source of vitamin C), d-calcium pantothenate, thiamine mononitrate, beta-carotene, riboflavin, pyridoxine hydrochloride, folic acid, biotin, vitamin B12 supplement), minerals (zinc proteinate, iron proteinate, copper proteinate, zinc oxide, manganese proteinate, copper sulphate, ferrous sulphate, calcium iodate, manganous oxide, selenium yeast), DL-methionine, glucosamine hydrochloride, chondroitin sulphate, yeast extract, yucca schidigera extract, dried rosemary. ^2^ Calculated nitrogen free extract. ^3^ Calculated metabolizable energy based on modified Atwater values.

**Table 2 animals-11-02603-t002:** Fatty acid profile of camelina oil, canola oil, flax oil, and sunflower oil.

Parameter	Sunflower ^1^	Canola ^2^	Flax ^2^	Camelina ^2^
Saturated Fatty Acids (%)	9.61	6.50	8.20	9.50
Monounsaturated Fatty Acids (%)	14.1	63.8	16.6	35.2
Polyunsaturated Fatty Acids (%)	76.3	29.7	75.2	55.3
Omega 6 (%)	76.2	18.6	16.5	19.8
Omega 3 (%)	0.04	11.1	58.6	35.4
Trans fat (%)	N/A ^2^	< 0.1	< 0.1	< 0.1
Total Fat (%)	N/A ^2^	99.9	100	99.9

^1^ Numerical values are adapted from Kostik et al. (2013) and only represent generic sunflower oil and not the specific brand used for this study [23]. ^2^ Samples run in duplicate by SGS Canada Inc., average values reported. ^3^ Abbreviation: N/A, Not Available.

**Table 3 animals-11-02603-t003:** Daily food intake for healthy, adult dogs fed one of three dietary oil supplements (camelina, flax, or canola oil) over a 16-week period. Values shown are lsmeans estimates for treatment (oil), week, and treatment by week interaction.

Food Intake Means Estimate (g/kg/Day)					*p*-Values		
Mean Category	Camelina	Canola	Flax	Mean ± SEM (Week)	P (Trmt)	P (Week)	P (Trmt × Week)
Trmt Mean ± SEM ^1^	12.24 ± 0.697	11.11 ± 0.697	11.27 ± 0.698		0.4733	<0.0001	0.8420
Wk 0- Wk 2	12.44	11.86	11.98	12.01 ^AB^ ± 0.382			
Wk 2-Wk 4	13.00	11.58	11.82	12.13 ^A^ ± 0.403			
Wk 4- Wk 6	12.98	11.46	11.66	12.04 ^AB^ ± 0.414			
Wk 6- Wk 8	11.96	11.11	11.23	11.55 ^AB^ ± 0.420			
Wk 8- Wk 10	12.30	11.00	11.36	11.43 ^ABC^ ± 0.467			
Wk 10- Wk 12	12.27	10.77	10.99	11.34 ^BC^ ± 0.458			
Wk 12- Wk 14	11.78	10.75	10.76	11.10 ^C^ ± 0.438			
Wk 14- Wk 16	11.23	10.37	10.38	10.66 ^D^ ± 0.396			

^1^ Abbreviation: SEM, Standard error of means; Wk, Week. Different superscripts used to reflect differences among weeks (*p* < 0.05).

**Table 4 animals-11-02603-t004:** Hematology of healthy, adult dogs fed one of three dietary oil supplements (camelina, flax, or canola oil) over a 16-week period. Values are shown are lsmeans estimates for treatment (oil), treatment by week interaction, and week.

	Treatment		Treatment × Week				Week (Pooled)				*p*-Values		
Marker	Ref. Range ^1^	Mean ± SEM ^1^	Pre-study	Wk ^1^ 4	Wk 10	Wk 16	Pre-study ± SEM	Wk 4 ± SEM	Wk 10 ± SEM	Wk 16 ± SEM	Trmt ^1^	Week	Trmt × Week
WBC ^1^ (×10^9^/L)	4.9–15.4						8.34 ± 0.512	8.46 ± 0.514	8.09 ± 0.515	8.25 ± 0.520	0.4025	0.8205	0.6789
Camelina		9.17 ± 0.837	9.79	9.01	8.67	9.22							
Canola		7.54 ± 0.802	7.09	8.05	7.42	7.61							
Flax		8.15 ± 0.787	8.13	8.33	8.19	7.93							
RBC ^1^ (×10^12^/L)	5.8–8.5						6.77 ± 0.130	6.77 ± 0.131	6.89 ± 0.131	6.86 ± 0.134	0.6592	0.5321	0.4282
Camelina		6.75 ± 0.209	6.57	6.74	6.73	6.94							
Canola		6.75 ± 0.200	6.69	6.79	6.82	6.70							
Flax		6.97 ± 0.197	7.04	6.77	7.13	6.95							
Hb ^1^ (g/L)	133–197						161 ± 3.056	162 ± 3.068	163 ± 3.071	166 ± 3.136	0.6249	0.1946	0.6497
Camelina		160 ± 4.936	156	160	158	167							
Canola		163 ± 4.730	160	163	163	164							
Flax		166 ± 4.655	166	163	168	168							
HCT ^1^ (L/L)	0.39–0.56						0.48 ± 0.010	0.48 ± 0.010	0.49 ± 0.010	0.48 ± 0.010	0.7855	0.6143	0.4396
Camelina		0.48 ± 0.015	0.47	0.49	0.48	0.48							
Canola		0.48 ± 0.014	0.47	0.48	0.49	0.48							
Flax		0.49 ± 0.014	0.50	0.48	0.51	0.48							
MCV ^1^ (fL)	66–75						71.2 ± 0.694	71.8 ± 0.699	71.0 ± 0.702	70.0 ± 0.737	0.5930	0.3554	0.8199
Camelina		70.9 ± 0.749	71.3	72.5	70.7	69.2							
Canola		71.5 ± 0.720	71.2	71.5	71.5	71.9							
Flax		70.5 ± 0.735	71.1	71.4	70.7	68.7							
MCH ^1^ (pg)	21–25						24.0 ^AB^ ± 0.191	24.1 ^AB^ ± 0.192	23.7 ^B^ ± 0.192	24.3 ^A^ ± 0.200	0.5001	0.0121	0.5807
Camelina		23.8 ± 0.289	23.9	23.8	23.3	24.0							
Canola		24.3 ± 0.277	24.1	24.2	24.1	24.6							
Flax		24.0 ± 0.274	23.7	24.4	23.7	24.3							
MCHC ^1^ (gL)	321–360						336 ± 3.919	335 ± 3.952	334 ± 3.961	348 ± 4.163	0.8247	0.0704	0.7874
Camelina		336 ± 4.051	335	327	333	350							
Canola		339 ± 3.895	341	338	335	343							
Flax		339 ± 3.994	333	339	334	351							
RDW ^1^ (%)	11–14						12.9 ± 0.275	12.9 ± 0.278	13.0 ± 0.277	13.6 ± 0.292	0.4994	0.3172	0.5112
Camelina		13.0 ± 0.253	12.8	12.3	13.2	13.8							
Canola		13.3 ± 0.243	13.2	13.4	13.5	13.2							
Flax		12.9 ± 0.252	12.7	13.0	12.3	13.7							
Platelets (×10^9^/L)	117–418						242 ± 20.97	254 ± 21.09	191 ± 21.18	185 ± 22.23	0.5852	0.0626	0.7476
Camelina		224 ± 24.51	270	265	194	166							
Canola		232 ± 23.55	234	240	227	226							
Flax		198 ± 23.90	220	257	151	163							
MPV ^1^ (fL)	7–14						10.6 ± 0.376	10.8 ± 0.379	10.6 ± 0.379	11.2 ± 0.397	0.2173	0.5714	0.7430
Camelina		10.8 ± 0.456	10.8	10.5	11.1	11.0							
Canola		11.3 ± 0.438	11.3	11.5	10.6	11.9							
Flax		10.2 ± 0.442	9.53	10.6	10.6	10.5							
Plateletcrit (%)	0.14–0.47						0.23 ^AB^ ± 0.020	0.27 ^A^ ± 0.021	0.19 ^B^ ± 0.021	0.20 ^AB^ ± 0.022	0.4580	0.0169	0.5971
Camelina		0.22 ± 0.022	0.24	0.26	0.19	0.18							
Canola		0.25 ± 0.021	0.24	0.26	0.23	0.25							
Flax		0.21 ± 0.022	0.21	0.30	0.14	0.18							
T.S. Protein ^1^ (g/L)	55–75						74.8 ^A^ ± 1.124	76.6 ^B^ ± 1.121	76.2 ^AB^ ± 1.128	78.0 ^AB^ ± 1.179	0.5546	0.0002	0.8743
Camelina		77.6 ± 1.571	75.3	77.7	77.7	80.2							
Canola		76.1 ± 1.508	74.3	76.8	76.2	76.9							
Flax		75.4 ± 1.494	74.7	75.3	74.7	77.0							
Seg. Neut. Count^1^ (×10^9^/L)	2.9–10.6						5.13 ± 0.339	5.20 ± 0.340	5.32 ± 0.340	5.33 ± 0.344	0.1307	0.8461	0.8341
Camelina		6.02 ± 0.556	6.08	6.02	5.82	6.17							
Canola		4.35 ± 0.533	3.99	4.22	4.64	4.56							
Flax		5.36 ± 0.523	5.31	5.37	5.50	5.25							
Lymphocyte Count (×10^9^/L)	0.8–5.1						2.13 ^A^ ± 0.131	1.97 ^AB^ ± 0.132	1.66 ^B^ ± 0.132	1.73 ^B^ ± 0.135	0.6318	0.0076	0.3049
Camelina		2.01 ± 0.180	2.34	2.03	1.66	2.01							
Canola		1.82 ±0.173	2.26	1.81	1.69	1.50							
Flax		1.79 ± 0.171	1.78	2.05	1.63	1.69							
Monocyte Count (×10^9^/L)	0.0–1.1						0.42 ± 0.054	0.35 ± 0.054	0.40 ± 0.054	0.41 ± 0.056	0.6238	0.5686	0.7791
Camelina		0.36 ± 0.066	0.45	0.30	0.39	0.29							
Canola		0.45 ± 0.063	0.44	0.40	0.41	0.53							
Flax		0.38 ± 0.063	0.36	0.34	0.40	0.42							
Eosinophil Count (×10^9^/L)	0.08–1.33						0.62 ± 0.115	0.64 ± 0.114	0.68 ± 0.114	0.66 ± 0.117	0.5960	0.9441	0.7665
Camelina		0.79 ± 0.175	0.82	0.83	0.75	0.79							
Canola		0.57 ± 0.166	0.40	0.55	0.64	0.69							
Flax		0.58 ± 0.166	0.63	0.54	0.64	0.52							

^1^ Abbreviations: SEM, standard error of means; Ref. Range, reference range; Wk, week; Trmt, treatment; RBC, erythrocyte count; WBC, white blood cell count; Hb, hemoglobin; Hct, hematocrit; MCV, mean cell volume; MCH, mean cell hemoglobin; MCHC, mean corpuscular hemoglobin concentration; RDW, red cell distribution width; MPV, mean platelet volume; T.S. protein, total solids protein; Seg. Neut. Count, segmented neutrophil count. ^A–C^ Values in a row and/or category with different superscripts are significantly different (*p* < 0.05).

**Table 5 animals-11-02603-t005:** Serum biochemistry for healthy, adult dogs fed one of three dietary oil supplements (camelina, flax, or canola oil) over a 16-week period. Values are shown are lsmeans estimates for treatment (oil), treatment by week interaction, and week.

	Treatment		Treatment × Week				Week (Pooled)				*p*-Values		
Marker	Ref. Range ^1^	Mean ± SEM ^1^	Pre-study	Wk ^1^ 4	Wk 10	Wk 16	Pre-study ± SEM	Wk 4 ± SEM	Wk 10 ± SEM	Wk 16 ± SEM	Trmt ^1^	Week	Trmt × Week
Calcium (mmol/L)	2.50–3.00						2.52 ± 0.024	2.55 ± 0.024	2.55 ± 0.025	2.58 ± 0.025	0.7484	0.2212	0.4107
Camelina		2.57 ± 0.032	2.55	2.61	2.56	2.55							
Canola		2.54 ± 0.031	2.52	2.51	2.53	2.59							
Flax		2.54 ± 0.031	2.49	2.52	2.55	2.59							
Phosphorus (mmol/L)	0.90–1.85						1.24 ± 0.043	1.16 ± 0.043	1.18 ± 0.044	1.22 ± 0.044	0.3729	0.1728	0.2832
Camelina		1.13 ± 0.061	1.26	1.07	1.09	1.10							
Canola		1.23 ± 0.058	1.26	1.20	1.20	1.26							
Flax		1.24 ± 0.570	1.19	1.22	1.25	1.32							
Sodium (mmol/L)	140–154						146 ^AB^ ± 0.324	147 ^A^ ± 0.324	145 ^B^ ± 0.331	146 ^A^ ± 0.340	0.5555	<0.0001	0.9041
Camelina		146 ± 0.408	146	147	146	147							
Canola		146 ± 0.395	146	146	145	147							
Flax		146 ± 0.391	146	147	145	146							
Magnesium (mmol/L)	0.7–1.0						0.84 ± 0.013	0.82 ± 0.013	0.83 ± 0.013	0.82 ± 0.014	0.4130	0.2481	0.7377
Camelina		0.82 ± 0.021	0.83	0.81	0.84	0.81							
Canola		0.85 ± 0.020	0.87	0.84	0.84	0.84							
Flax		0.81 ± 0.020	0.82	0.81	0.82	0.81							
Potassium (mmol/L)	3.8–5.4						4.99 ^A^ ± 0.058	4.90 ^AB^ ± 0.058	4.70 ^C^ ± 0.059	4.78 ^BC^ ± 0.060	0.5878	0.0002	0.7987
Camelina		4.81 ± 0.077	5.03	4.84	4.61	4.78							
Canola		4.91 ± 0.074	5.04	4.93	4.81	4.86							
Flax		4.81 ± 0.073	4.91	4.92	4.70	4.70							
Chloride (mmol/L)	104–119						111 ^A^ ± 0.465	110 ^AB^ ± 0.465	109 ^B^ ± 0.471	110 ^AB^ ± 0.478	0.5778	0.0045	0.1752
Camelina		111 ± 0.747	111	111	111	111							
Canola		110 ± 0.720	111	110	108	110							
Flax		110 ± 0.703	111	110	109	110							
Carbon Dioxide (mmol/L)	15–25						18.1 ^A^ ± 0.294	16.6 ^B^ ± 0.298	17.4 ^AB^ ± 0.306	17.6 ^AB^ ± 0.326	0.8910	0.0025	0.6793
Camelina		17.4 ± 0.301	18.0	16.8	17.2	17.4							
Canola		17.4 ± 0.284	18.5	16.1	17.5	17.3							
Flax		17.5 ± 0.281	17.7	16.8	17.5	18.0							
Anion Gap (mmol/L)	13–24						22.1 ^B^ ± 0.459	24.8 ^A^ ± 0.463	23.4 ^B^ ± 0.473	23.4 ^AB^ ± 0.501	0.7104	<0.0001	0.6028
Camelina		23.4 ± 0.628	22.6	24.2	23.2	23.4							
Canola		23.8 ± 0.596	21.8	25.5	24.0	23.9							
Flax		23.1 ± 0.589	21.8	24.7	22.9	23.0							
Na:K Ratio ^1^	29–37						29.5 ^B^ ± 0.375	30.2 ^AB^ ± 0.375	31.0 ^A^ ± 0.383	31.0 ^A^ ± 0.391	0.4702	0.0015	0.6193
Camelina		30.7 ± 0.506	29.4	30.6	32.0	31.0							
Canola		29.9 ± 0.489	29.2	29.8	30.0	30.5							
Flax		30.6 ± 0.481	29.8	30.2	30.9	31.4							
Total Protein (g/L)	55–74						62.2 AB ± 0.786	62.9 A ± 0.790	61.2 B ± 0.795	61.5 AB ± 0.798	0.5071	0.0078	0.2894
Camelina		63.2 ± 1.345	63.2	64.6	62.1	63.0							
Canola		61.4 ± 1.285	61.0	62.6	61.3	60.8							
Flax		61.2 ± 1.259	62.4	61.5	60.1	60.8							
Albumin (g/L)	29–43						37.5 ^AB^ ± 0.436	37.3 ^B^ ± 0.436	37.1 ^B^ ± 0.439	38.0 ^A^ ± 0.442	0.9721	0.0087	0.2812
Camelina		37.6 ± 0.747	37.1	37.4	37.3	38.5							
Canola		37.3 ± 0.720	37.9	36.9	36.8	37.8							
Flax		37.5 ± 0.700	37.6	37.6	37.2	37.8							
Globulin (g/L)	21–42						24.2 ^AB^ ± 0.799	25.6 ^A^ ± 0.803	24.1 ^AB^ ± 0.810	23.5 ^B^ ± 0.814	0.6714	0.0058	0.1424
Camelina		25.3 ± 1.335	24.7	27.1	24.8	24.6							
Canola		24.1 ± 1.275	23.1	25.7	24.5	22.9							
Flax		23.7 ± 1.250	24.8	23.9	23.0	23.1							
A:G Ratio^1^	0.7–1.8						1.57 ^AB^ ± 0.053	1.51 ^B^ ± 0.053	1.61 ^AB^ ± 0.054	1.67 ^A^ ± 0.054	0.7507	0.0049	0.1078
Camelina		1.54 ± 0.085	1.49	1.44	1.59	1.63							
Canola		1.62 ± 0.081	1.70	1.48	1.58	1.73							
Flax		1.61 ± 0.079	1.53	1.60	1.66	1.66							
Urea (mmol/L)	3.5–9.0						5.62 ^A^ ± 0.244	4.92 ^B^ ± 0.247	4.83 ^B^ ± 0.252	4.96 ^AB^ ± 0.258	0.0687	0.0129	0.6221
Camelina		4.96 ± 0.347	5.65	5.06	4.66	4.45							
Canola		5.72 ± 0.333	6.21	5.42	5.28	5.98							
Flax		4.57 ± 0.327	5.01	4.27	4.56	4.44							
Creatine (μmol/L)	20–150						78.3 ± 3.971	82.9 ± 3.989	80.2 ± 4.017	82.1 ± 4.091	0.4840	0.2917	0.3730
Camelina		82.9 ± 6.805	76.7	86.3	81.0	87.5							
Canola		85.1 ± 6.496	82.9	86.8	83.0	87.7							
Flax		74.6 ± 6.356	75.1	75.5	76.8	71.0							
Glucose (mmol/L)	3.3–7.3						5.14 ^B^ ± 0.079	5.32 ^AB^ ± 0.079	5.21 ^AB^ ± 0.081	5.39 ^A^ ± 0.083	0.1811	0.0245	0.6345
Camelina		5.22 ± 0.115	5.15	5.33	5.22	5.20							
Canola		5.14 ± 0.111	5.05	5.17	5.04	5.30							
Flax		5.43 ± 0.109	5.24	5.46	5.37	5.65							
Total Bilirubin (μmol/L)	0–4						0.90 ^B^ ± 0.138	1.34 ^A^ ± 0.138	1.46 ^A^ ± 0.141	1.30 ^A^ ± 0.142	0.7096	0.0006	0.4048
Camelina		1.39 ± 0.205	0.93	1.45	1.52	1.65							
Canola		1.20 ± 0.197	0.72	1.32	1.48	1.27							
Flax		1.17 ± 0.193	1.05	1.25	1.39	0.99							
Conj. Bilirubin^1^ (μmol/L)	0–1						0.53 ^B^ ± 0.081	0.73 ^AB^ ± 0.081	0.76 ^A^ ± 0.082	0.69 ^AB^ ± 0.084	0.9948	0.0354	0.0208
Camelina		0.68 ± 0.115	0.31 ^B^	0.90 ^A^	0.80 ^AB^	0.69 ^AB^							
Canola		0.67 ± 0.111	0.54 ^AB^	0.54 ^AB^	0.84 ^AB^	0.76 ^AB^							
Flax		0.69 ± 0.109	0.73 ^AB^	0.73 ^AB^	0.65 ^AB^	0.63 ^AB^							
Free Bilirubin (μmol/L)	0–3						0.41 ± 0.124	0.64 ± 0.124	0.73 ± 0.127	0.63 ± 0.128	0.8368	0.1050	0.1666
Camelina		0.67 ± 0.175	0.64	0.52	0.59	0.93							
Canola		0.60 ± 0.169	0.27	0.78	0.74	0.62							
Flax		0.53 ± 0.166	0.31	0.61	0.84	0.34							

Cholesterol (mmol/L)	3.60–10.20						6.46 ± 0.361	6.61 ± 0.361	6.44 ± 0.362	6.44 ± 0.365	0.8759	0.3339	0.0011
Camelina		6.58 ± 0.645	6.38 ^ABCD^	6.83 ^ABCD^	6.42 ^ABCD^	6.67 ^ABCD^							
Canola		6.23 ± 0.619	5.81 ^ABC^	6.22 ^ABCD^	6.54 ^ABD^	6.33 ^ABCD^							
Flax		6.66 ± 0.604	7.20 ^ACD^	6.76 ^ABCD^	6.37 ^BCD^	6.32 ^ABCD^							
ALP^1^ (U/L)	22–143						44.8 ^B^ ± 6.875	67.2 ^A^ ± 10.29	56.6 ^A^ ± 8.702	61.4 ^A^ ± 9.496	0.4755	<0.0001	0.6648
Camelina		72.7 ± 19.65	54.6	83.7	70.3	87.1							
Canola		44.9 ± 11.70	37.6	50.1	47.5	45.6							
Flax		56.3 ± 14.25	44.0	72.3	54.1	58.4							
Steroid-Ind. ALP^1^ (U/L)	0–84						11.8 ^B^ ± 3.547	19.2 ^A^ ± 5.919	13.5 ^AB^ ± 4.110	15.2 ^AB^ ± 4.67	0.2262	0.0112	0.8550
Camelina		26.6 ± 13.52	20.2	33.4	25.4	29.0							
Canola		6.49 ± 3.796	6.21	7.77	5.64	6.51							
Flax		18.4 ± 8.687	13.1	27.3	17.1	18.7							
GGT^1^ (U/L)	0–7						1.34 ± 0.343	1.93 ± 0.343	1.58 ± 0.357	0.98 ± 0.362	0.0130	0.2539	0.1589
Camelina		2.08 ^A^ ± 0.352	0.81	2.96	2.44	2.13							
Canola		0.54 ^B^ ± 0.339	1.27	0.29	0.67	0.00							
Flax		1.75 ^A^ ± 0.338	1.94	2.54	1.63	0.90							
ALT ^1^ (U/L)	19–107						40.0 ^B^ ± 3.602	49.0 ^A^ ±4.414	51.4 ^A^ ± 4.700	52.2 ^A^ ± 4.799	0.0209	0.0005	0.3563
Camelina		58.6 ^A^ ± 8.749	43.3	58.1	70.2	66.8							
Canola		55.1 ^AB^ ± 7.903	48.4	55.4	57.7	59.7							
Flax		33.9 ^B^ ± 4.751	30.4	36.4	33.4	35.7							
CK ^1^ (U/L)	40–255						92.2 ^B^ ± 8.414	108 ^AB^ ± 9.817	119 ^AB^ ± 11.22	126 ^A^ ± 12.03	0.2677	0.0290	0.8641
Camelina		94.2 ± 11.22	81.6	82.9	108	108							
Canola		123 ± 14.17	109	124	120	144							
Flax		116 ± 13.09	88.1	121	129	130							
Amylase (U/L)	299–947						638 ^B^ ± 47.85	688 ^A^ ± 47.85	688 ^AB^ ± 48.14	679 ^AB^ ± 48.49	0.6248	0.0410	0.4374
Camelina		672 ± 83.99	643	654	698	693							
Canola		730 ± 80.56	670	760	739	749							
Flax		618 ± 78.70	601	649	628	594							
Lipase (U/L)	25–353						107 ^A^ ± 9.935	87.3 ^B^ ± 8.097	101 ^AB^ ± 9.589	89.8 ^AB^ ± 8.691	0.0722	0.0182	0.5192
Camelina		106 ± 14.04	113	84.1	124	108							
Canola		75.3 ± 9.418	83.4	67.4	75.0	72.4							
Flax		112 ± 13.97	130	117	110	92.4							
Calc. Osmo. ^1^ (mmol/L)	N/A ^1^						292 ^A^ ± 0.706	292 ^A^ ± 0.713	289 ^B^ ± 0.728	291 ^A^ ± 0.754	0.6718	<0.0001	0.8177
Camelina		291 ± 0.934	293	293	289	291							
Canola		291 ± 0.895	292	292	288	293							
Flax		290 ± 0.885	290	292	288	291							

^1^ Abbreviations: SEM, standard error of means; Ref. Range, reference range; Wk, week; Trmt, treatment; Na:K Ratio, sodium;potassium ratio; A:G Ratio, albumin:globulin ration; Conj. Bilirubin, conjugated bilirubin; ALP; alkaline phosphatase; Steroid-Ind. ALP, steroid-induced alkaline phosphatase; GGT, gamma-glutamyl transferase; ALT, alanine aminotrasnferase; CK, creatine kinase; Calc. Osmo., calculated osmolarity; N/A, not available. ^A–C^ Values in a row and/or category with different superscripts are significantly different (*p* < 0.05).

## Data Availability

The data presented in this study are available on request from the corresponding author.

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
