# Peer review of "Safety of Dietary Camelina Oil Supplementation in Healthy, Adult Dogs"

_animals, 2021, doi:10.3390/ani11092603_

Round 1

Reviewer 1 Report

This is a very interesting and important study. 

The main claim of the paper is that camelina oil could be fed to dogs without any negative effects on the "body weight, body condition score, food intake, or hematology and biochemistry analytes" measured in the study. This information is important in considering fatty acid sources that can be safely added to canine diets, especially as plant-based sources of fatty acids become increasingly favored for many reasons. My search of the literature determined this to be a novel study and I would not see need for additional information to support the claims of the study.

Author Response

Thank you for your review and support for this publication.

Reviewer 2 Report

Line 25: Unity is missing - …. (BW) of 27,4 ??

Table 1: Metabolizable Energy – KJ or MJ instead of kcal

Line 46 to 49: Please check the data

 Line 110: Unity is missing - …….(110 ?? x BW ?? 0,75)

Table 4: Characteristics: T.S.Protein – Request for control of uppercase letters

Table 5: Characteristics: Cholesterol and Conjugated Bilirubin concerning treatment - Request for control of uppercase letters

Author Response

Thank you for the review on this paper, it is greatly appreciated. 

Minor edits have been made accordingly for Line 25 and Line 110. 

Line 46 to 49: Please check the data

  • This is data has been converted to a more reader friendly n-6:n-3 ratio formatting.

Table 1: Metabolizable Energy – KJ or MJ instead of kcal:

  • Thank you for this suggestion. We noted that other manuscript in Animals use kcal and this is the way in which dietary energy is reported for dog food in North America. Due to the commercial relevance to North America, we feel that we should keep the units in kcal.

Table 4: Characteristics: T.S.Protein – Request for control of uppercase letters

  • We fixed the uppercase in the footnote of the table to match T.S. Protein shown in the table if that is what is meant, though based on the comments regarding Table 5 I am not sure.

Table 5: Characteristics: Cholesterol and Conjugated Bilirubin concerning treatment - Request for control of uppercase letters

  • Thank you for this comment, however, we are not completely sure of the meaning. Cholesterol and Conjugated Bilirubin have the same capitalization formatting as the other biomarkers listed in Table 5, and while Cholesterol and Conjugated Bilirubin do use uppercase letters as superscripts to denote significance, many other biomarkers do as well and were not commented on so that may not be the intended meaning. Are you able to provide further clarification as to what changes we should be making?

Thank you again for your feedback and if there are any further changes you would recommend, please let us know. 

Reviewer 3 Report

I'd like to congratulate the Authors on the great manuscript.

My only remark is that there are typos in some Tables footnotes, that need to be corrected in lines 226, 242-243.

I'd like to ask would you consider studying further the health promoting properties of the oil? Recent example are given in the recent text linked below:

https://www.sciencedirect.com/science/article/pii/B9780128181881000219

The detailed fatty acid profile (of the oil) would be indeed interesting to know.

Author Response

We would like to thank you for taking the time to review this manuscript and provide feedback, it is greatly appreciated.

Typos that were identified have been corrected appropriately. Thank you for bringing attention to these.

Analyses are underway to examine the effects of feeding these oils on trans-epidermal water loss, skin and coat characteristics, pro-and anti-inflammatory biomarkers, and serum lipid and fatty acid profiles (as well as detailed oil profiles). These results will likely be published early 2022. 

Thank you again for your feedback on this manuscript.